# Benefits of Switching Mycophenolic Acid to Sirolimus on Serological Response after a SARS-CoV-2 Booster Dose among Kidney Transplant Recipients: A Pilot Study

**DOI:** 10.3390/vaccines10101685

**Published:** 2022-10-09

**Authors:** Athiphat Banjongjit, Supitchaya Phirom, Jeerath Phannajit, Watsamon Jantarabenjakul, Leilani Paitoonpong, Wonngarm Kittanamongkolchai, Salin Wattanatorn, Wisit Prasithsirikul, Somchai Eiam-Ong, Yingyos Avihingsanon, Pokrath Hansasuta, Jakapat Vanichanan, Natavudh Townamchai

**Affiliations:** 1Division of Nephrology, Department of Medicine, Faculty of Medicine, Chulalongkorn University and King Chulalongkorn Memorial Hospital, Bangkok 10330, Thailand; 2Division of Clinical Epidemiology, Department of Medicine, Faculty of Medicine, Chulalongkorn University, Bangkok 10330, Thailand; 3Thai Red Cross Emerging Infectious Diseases Clinical Center, King Chulalongkorn Memorial Hospital, Bangkok 10330, Thailand; 4Department of Pediatrics, Faculty of Medicine, Chulalongkorn University, Bangkok 10330, Thailand; 5Division of Infectious Diseases, Department of Medicine, Faculty of Medicine, Chulalongkorn University, Bangkok 10330, Thailand; 6Mahachakri Sirindhorn Clinical Research Center, Faculty of Medicine, Chulalongkorn University, Bangkok 10330, Thailand; 7Renal Immunology and Renal Transplant Research Unit, Department of Medicine, Faculty of Medicine, Chulalongkorn University, Bangkok 10330, Thailand; 8Bamrasnaradura Infectious Disease Institute, Nonthaburi 11000, Thailand; 9Department of Microbiology, Faculty of Medicine, Chulalongkorn University and King Chulalongkorn Memorial Hospital, Bangkok 10330, Thailand; 10Excellence Center for Solid Organ Transplantation, King Chulalongkorn Memorial Hospital, Bangkok 10330, Thailand

**Keywords:** COVID-19 vaccination, AZD1222, BNT162b2, kidney transplantation, immunosuppression, mycophenolic acid sparing, CNI reduction, anti-SARS-CoV-2 S antibody

## Abstract

Kidney transplant recipients (KTRs) have a suboptimal immune response to COVID-19 vaccination due to the effects of immunosuppression, mostly mycophenolic acid (MPA). This study investigated the benefits of switching from the standard immunosuppressive regimen (tacrolimus (TAC), MPA, and prednisolone) to a regimen of mammalian target of rapamycin inhibitor (mTORi), TAC and prednisolone two weeks pre- and two weeks post-BNT162b2 booster vaccination. A single-center, opened-label pilot study was conducted in KTRs, who received two doses of ChAdOx-1 and a single dose of BNT162b2. The participants were randomly assigned to continue the standard regimen (control group, n = 14) or switched to a sirolimus (an mTORi), TAC, and prednisolone (switching group, n = 14) regimen two weeks before and two weeks after receiving a booster dose of BNT162b2. The anti-SARS-CoV-2 S antibody level after vaccination in the switching group was significantly greater than the control group (4051.0 [IQR 3142.0–6466.0] BAU/mL vs. 2081.0 [IQR 1077.0–3960.0] BAU/mL, respectively; *p* = 0.01). One participant who was initially seronegative in the control group remained seronegative after the booster dose. These findings suggest humoral immune response benefits of switching the standard immunosuppressive regimen to the regimen of mTORi, TAC, and prednisolone in KTRs during vaccination.

## 1. Introduction

Coronavirus disease 2019 (COVID-19) has a high mortality rate among solid organ transplant recipients (SOTRs). The mortality rate has decreased over time among kidney transplant recipients (KTRs). The mortality rate during March 2020 to February 2021 was 34.6% among KTRs. However, after the widespread use of Severe Acute Respiratory Syndrome Coronavirus 2 (SARS-CoV-2) vaccines, the mortality rate among KTRs declined to 13.3% [1]. Nevertheless, the infection rate is still considered to be high among KTRs. In our center, the infection rate of COVID-19 among KTRs was 12.8% during January to June 2022 and 7% of them developed pneumonia. The immune response to vaccination in transplant recipient is poor compared to the general population [2]. Since the efficacy of the vaccination among SOTRs was low, the Centers for Disease Control and Prevention (CDC) recommended that SOTRs should receive a three-dose primary series of messenger Ribonucleic Acid (mRNA) vaccine with at least one booster dose, which is more than healthy people, who require only a two-dose primary series of mRNA vaccine with at least one booster dose [3]. The reason why SOTRs should receive more doses of the vaccine is due to the immunosuppressive regimen used to prevent graft rejection after transplantation. The standard immunosuppressive regimen consists of tacrolimus (TAC), mycophenolic acid (MPA), and prednisolone [4,5]. A regimen of mammalian target of rapamycin inhibitor (mTORi), TAC and prednisolone provides comparable outcomes with the standard regimen [6,7,8]. MPA is one of the most important factors that impairs the immune response [9,10,11,12,13,14,15]. Hence, MPA dose pause during vaccination is one of the strategies to improve immune response [16]. However, the short-term MPA pause may lead to inadequate immunosuppression during vaccination and the long-term result remains unanswered. Recent studies that compared the immune response after COVID-19 vaccination among patients who received the standard immunosuppressive regimen had poorer immune response compared to those on mTORi, TAC and prednisolone; patients on mTORi, TAC, and prednisolone had higher cellular and humoral immunity after vaccination [17,18]. Herein, we evaluated a strategy to improve the immune response to COVID-19 vaccination among KTRs who were on the standard immunosuppressive regimen by switching the immunosuppressive regimen to the mTORi, TAC, and prednisolone regimen to ensure that KTRs had adequate immunosuppression during vaccination.

In the present pilot study, we compared the immune response to the booster dose of BNT162b2 in KTRs who were on the standard immunosuppressive regimen and those who switched to the mTORi, low-dose TAC, and prednisolone regimen.

## 2. Materials and Methods

### 2.1. Study Design

This is a single-center, pilot, open-label study conducted at the King Chulalongkorn Memorial Hospital, Bangkok, Thailand, from March to September 2022. Three months after receiving the BNT162b2 dose, we administered a booster dose (adapted from CDC recommendation [3]) of homologous vaccine to KTRs in our center who mostly received two doses of AZD1222 and a single dose of BNT162b2. Participants who were older than 18 years old, undergoing kidney transplantation for more than 6 months with stable graft function, on a standard immunosuppressive regimen (TAC, MPA, and prednisolone), and had two doses of AZD1222 and a single dose of BNT162b2 for more than 3 months were enrolled into the study. Participants with a history of severe allergy to BNT162b2 and who required hospitalization, had a history of SARS-CoV-2 infection (had a positive SARS-CoV-2 polymerase chain reaction [PCR] result), had active rejection or infection within 3 months, had donor specific anti-HLA antibody (DSA), were pregnant or lactating, had metastatic malignancy, and human immunodeficiency virus infection were excluded from the study.

### 2.2. Immunosuppressive Regimens and Vaccination

The participants were enrolled into the study by the research assistant using a computer. The enrolled participants were randomly assigned in a 1:1 ratio using a block of four randomization to either continuing the standard immunosuppressive regimen (control group) or switching to the mTORi, TAC, and prednisolone regimen (switching group). The randomization was performed by an independent statistician. Sequentially numbered, opaque and sealed envelopes were used for the allocation concealment. It is not feasible to conduct a double-blinded study because there is a distinction in the drug characteristics. The standard immunosuppressive regimen consisted of TAC (Prograf^®^ or Advagraf^®^, Astellas, Tokyo, Japan) with a trough level of 4–7 ng/mL, MPA (mycophenolate mofetil; Cellcept^®^, Roche, Basel, Switzerland, 1000 to 1500 mg/day, or enteric coated-mycophenolate sodium; Myfortic^®^, Novartis, Basel, Switzerland, 720 to 1080 mg/day), and prednisolone. The immunosuppressive drugs in the switching group were mTORi (Sirolimus, Rapamune^®^, Pfizer, New York, NY, USA) with trough level of 5–10 ng/mL, low-dose TAC (Advagraf^®^, Astellas, Tokyo, Japan) with trough level of 2–4 ng/mL, and prednisolone. The switching of the immunosuppressive regimen was started two weeks before the BNT162b2 (Pfizer, New York, NY, USA) booster was administered to restore the immune response which had been suppressed by MPA, and continued for two more weeks after the booster vaccination to allow the immune response to build up against COVID-19 (a total of four weeks during this period). Sirolimus was administered as a loading dose using 6 mg in day 1 and henceforth followed by 2 mg daily. Low-dose tacrolimus was administered as a half dose of the previous dosage. After switching the drugs for 2 weeks, the trough levels of tacrolimus and sirolimus were measured to adjust the dosage of the drugs.

This study was approved by the Institutional Review Board of the Faculty of Medicine, Chulalongkorn University (IRB No.230/65). The study was conducted in compliance with the international guidelines for human research protection as per the Declaration of Helsinki and the International Conference on Harmonization in Good Clinical Practice (ICH-GCP). The study protocol was registered in the Thai clinical trial registration (TCTR20220404001). All participants provided written informed consent prior to their enrollment in the study and the medical records were thoroughly reviewed.

### 2.3. Outcome Measurements

Baseline characteristics and biochemical data of the participants were obtained from their electronic medical records. The mean of the previous three consecutive serum creatinine levels was used as the baseline serum creatinine for the participant. The outcome was a change in anti-SARS-CoV-2 S antibody level pre- and post-BNT162b2 vaccination. Blood samples were collected before vaccination and at 4 weeks after vaccination and centrifuged at 3000 rounds per minute for 10 min at room temperature and frozen at −20 °C until the anti-SARS-CoV-2 S antibody was measured (Elecsys^®^, by Cobas e 411 analyzer; Roche Diagnostics, Basel, Switzerland). The lower limit of detection was 0.36 binding antibody units (BAU)/mL. The anti-SARS-CoV-2 S antibody ≥0.823 BAU/mL was considered reactive or seroconverted. The adverse events were monitored for six months or until September 2022.

### 2.4. Statistical Analysis

Continuous data were presented as mean ± standard deviation (SD) for Gaussian distributed data and median (interquartile range, IQR) for non-Gaussian distributed data. Categorical data were described as proportion and percentages. Baseline characteristics were compared between the groups using t-test for normal distributed data and the Wilcoxon rank-sum test for non-normal distributed data. The serum antibodies which were seronegative were imputed with 0.18 BAU/mL (50% of the lower limit of detection). The difference in the increment of the anti-SARS-CoV-2 S antibody level between the two groups was compared by t-test. The data were analyzed using Stata 14 (StataCorp LP, College Station, TX, USA) and a *p*-value < 0.05 was considered statistically significant. The visualizations were performed using GraphPad Prism 9 (GraphPad Software, San Diego, CA, USA).

## 3. Results

### 3.1. Baseline Characteristics of the Participants

A total of 693 active KTRs in our center were screened. Thirty-one patients met the inclusion criteria. Three patients declined to participate in the study. A total of 28 participants were randomly assigned to the control group (14 participants) and the switching group (14 participants). All participants completed the study (Figure 1). The mean age (±SD) of the KTRs was 51.5 ± 8.7 years and was not different between the two groups. The median (IQR) transplant vintage of the control and the switching groups were 3.2 (IQR 1.5–14.4) years and 3.3 (IQR 1.8–7.3) years (*p* = 0.85), respectively. The duration since the last dose of BNT162b2 was comparable at approximately 5 months between both groups (*p* = 0.66). The white blood cells, neutrophil, and lymphocyte counts were also comparable between the two groups (Table 1). In the switching group, a trough level achieved 3.3 ± 1.6 ng/mL for TAC and 10.3 ± 4.0 ng/mL for sirolimus.

### 3.2. Post-Vaccination Anti-SARS-CoV-2 S Antibody Level and Seroconversion Rate

The median (IQR) baseline anti-SARS-CoV-2 S antibody level was 170.2 (IQR 36.0–510.3) BAU/mL, which was not different between the control group (204.9 [IQR 44.7–541.2] BAU/mL) and the switching group (164.5 [IQR 19.5–429.5] BAU/mL) (*p* = 0.61). After vaccination, the overall anti-SARS-CoV-2 S antibody level significantly increased to 3338.0 (IQR 2081.0–4983.5) BAU/mL (*p* < 0.001). The anti-SARS-CoV-2 S antibody level significantly increased in both groups (*p* = 0.001). The switching group had a significantly higher anti-SARS-CoV-2 S antibody level compared to the control group (4051.0 [IQR 3142.0–6466.0] vs. 2081.0 [IQR 1077.0–3960.0], *p* = 0.01) (Figure 2).

In this study, there was only one seronegative participant which was in the control group. The participant did not achieve seroconversion after the fourth vaccination.

### 3.3. Adverse Events

Two participants in the switching group developed mild oral ulcers that spontaneously resolved after changing the mTORi back to MPA at the end of the study. There were no other adverse events related to mTORi, such as edema and pneumonitis. In the control group, there were no serious local or systemic adverse events such as bleeding, bruises, chest discomfort, severe headache, vomiting, seizure, or stroke-like symptoms after vaccination (Table 2). There were no significant changes in serum creatinine (1.40 ± 0.59 vs. 1.40 ± 0.67 mg/dL; *p* = 0.87 in control group and 1.30 ± 0.48 vs. 1.32 ± 0.43 mg/dL; *p* = 0.50 in switching group) and urine protein (165 [IQR 0–300] vs. 160 [IQR 0–330] mg/day; *p* = 0.27 in control group and 59.5 [IQR 0–350] vs. 199 [IQR 0–267] mg/day; *p* = 0.75 in switching group) before and after the study, respectively.

## 4. Discussion

The results in the present study indicated that there was a higher immune response after the booster dose of COVID-19 vaccine in the group that received the MPA-sparing regimen with mTORi and low-dose TAC compared to the standard immunosuppressive regimen. The only one seronegative KTR who was in the control group remained seronegative after the fourth COVID-19 vaccination.

Many studies reported that SOTRs have lower immunological response to vaccination compared to the healthy population [2,19,20,21]. The blunted vaccine immune responses among SOTRs was due to the use of immunosuppression, predominantly the MPA [2,9,10,11,12,13,14,16]. Fifty-seven percent of SOTRs with antimetabolites (mycophenolic acid or azathioprine) had negative antibody response after two doses of mRNA vaccine compared to 18% of those without antimetabolites [2]. A recent meta-analysis in patients with rheumatic diseases revealed that using MPA reduced the rate of seroconversion after SARS-CoV-2 vaccination by 44% [22].

The regimen of TAC, MPA and prednisolone is the standard immunosuppressive regimen widely used in kidney transplantation. However, the regimen which contained MPA provided lower immune response to many vaccines, such as influenza [23], hepatitis A [24], and SARS-CoV-2 [9,10,11,12,13,14,17,18]. Novel strategies to improve the immune response by modifying the immunosuppressive regimen have been investigated [16]. MPA dose reduction or pausing provided better immune response compared to the patients without MPA dose changing [16]. However, MPA dose modification or pause may lead to inadequate immunosuppression [25,26]. In this regard, a CNI reduction regimen with mTORi, TAC, and prednisolone yielded comparable efficacy and lower infection rate compared to patients on the standard regimen [6,7,8]. Therefore, this regimen can serve as MPA sparing and effectively provide adequate immunosuppression.

In a subgroup vaccination study of an RCT by Boer et al., conducted in 32 patients (16 patients on TAC [5 to 8 ng/mL] + MMF [1000 mg/day] + prednisolone and 16 patients on low dose TAC [1.5 to 4 ng/mL] + everolimus [3 to 6 ng/mL] + prednisolone), it was revealed that the SARS-CoV-2 anti-spike receptor binding domain IgG antibody levels were significantly higher in the low dose TAC + everolimus + prednisolone group after receiving two doses of mRNA vaccine [18]. All patients in the study were randomized to this immunosuppressive regimen since kidney transplantation. In contrast, in our study, we utilized the switching strategy to improve the immune response in KTRs who were on the standard immunosuppressive regimen.

After the participants received the booster dose, the antibody level in the switching group was higher than those on the standard immunosuppressive regimen. This can be explained by two assumptions. Firstly, the cessation of MPA and use of low dose of TAC resulted in an improvement of the immune response. Secondly, mTORi may also augment the immune response. The poor immune response after vaccination in patients who were on MPA could be explained by MPA inhibiting IL-4+ CD4 T-cell and B-cell function, plasma cell formation, and antibody production [15,27]. Thus, interrupting MPA during the period of vaccination is assumed to restore the immune response. The mTOR inhibitor was assumed to promote the immune response but the mechanism is still unclear. Previous studies showed that mTOR has a major role in regulating memory CD8 T-cell differentiation. Studies in mice revealed that rapamycin, a specific inhibitor of mTOR, increased the magnitude of acute lymphocytic choriomeningitis virus (LCMV)-specific CD8 T-cell response, increased the functional qualities of the memory CD8 T-cells, and promoted memory CD4 T-cell differentiation [28,29]. mTOR blockage by rapamycin could boost Toll-like receptor (TLR)-induced antigen-specific T- and B-cell responses to HBV vaccines [30].

In this study, a four-week period of switching to an MPA-sparing immunosuppressive regimen (two weeks before and two weeks after vaccination) was sufficient enough to improve antibody response. A recent study of MPA dose pause for one week before and four weeks after vaccination also revealed improvement in serologic response [16]. In another study with a shorter period, two weeks of immunosuppressive drug modification revealed an inadequate immune response [31]. Since the immune system requires a certain amount of time to respond after suppression or cessation of immunosuppression [32], thus, a two-week duration of immunosuppressive regimen switching prior to vaccination will allow the immune system to be restored, and another two weeks after vaccination can help the body to build an immune response to vaccination. The strategy used in this study is beneficial for KTRs to develop an immune response after the vaccination.

The immune response to vaccination can be affected by various factors, such as the previous vaccination regimen and the duration after the last dose. The regimen of vaccination of all participants prior to entering this study was tightly controlled to be homogeneous, which was two doses of AZD1222 and a single dose of BNT162b2. Furthermore, the duration since the last vaccination was comparable between the two groups. Admittedly, this study only measured the anti-SARS-CoV-2 S antibody level. In this respect, it has been demonstrated that anti-SARS-CoV-2 S antibody level together, with neutralizing antibodies, could prevent the acquisition of SARS-CoV-2 infection and had a good correlation [33,34]. However, in this study, follow-up antibody tests were not performed so we do not know whether the benefits of immunosuppression switching persisted. It should be noted that the cellular immune response was not measured in our study. Asymptomatic infected participants might be not detected because we did not perform anti-nucleocapsid (N) antibody tests. This study was a pilot study with a small sample size and the participants in the study were not at high risk for graft rejection. The seroconversion rate could not be assessed since most of the participants were seropositive for anti-SARS-CoV-2 S antibody except for one participant. Additional studies with a larger sample size, with a higher immunologic risk and a longer follow-up period, are warranted to assess the anti-SARS-CoV-2 S antibody level, seroconversion rate, and vaccination efficacy.

## 5. Conclusions

Immunosuppressive drug switching from TAC, MPA, and prednisolone to the regimen of mTORi, TAC and prednisolone for two weeks before and two weeks after booster dose of COVID-19 vaccination can improve humoral response in stable KTRs.

## Figures and Tables

**Figure 1 vaccines-10-01685-f001:**
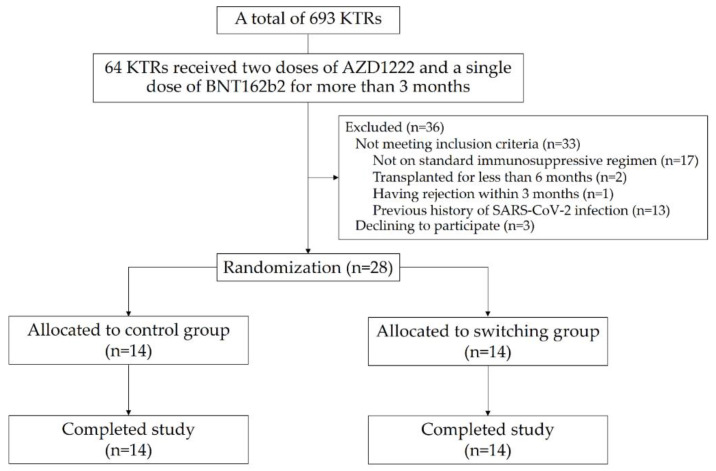
Flow diagram of the study participants.

**Figure 2 vaccines-10-01685-f002:**
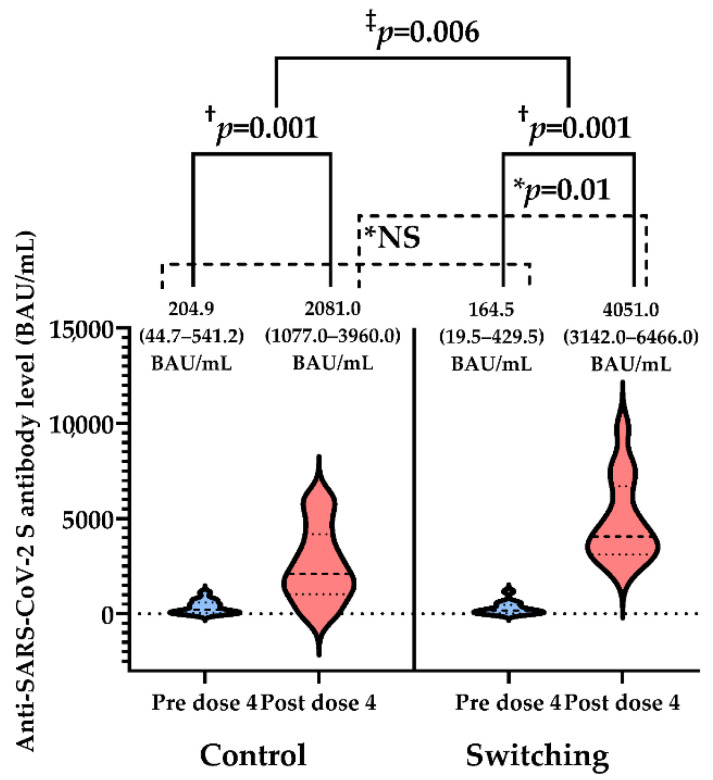
The anti-SARS-CoV-2 S antibody level (median [IQR]) pre- and post-fourth vaccination with BNT162b2 between the control group and the switching group. Baseline anti-SARS-CoV-2 S antibody level was not different between both groups (* *p* > 0.05, NS [not significant]). After vaccination with BNT162b2, anti-SARS-CoV-2 S antibody level increased in both groups († *p* = 0.001). Anti-SARS-CoV-2 S antibody level post-fourth vaccination with BNT162b2 in the switching group was significantly higher than the control group (* *p* = 0.01). The difference-in-differences was also significant (‡ *p* = 0.006). The horizontal dotted line indicates the cutoff point at 0.823 BAU/mL.

**Table 1 vaccines-10-01685-t001:** Baseline characteristics of the study participants.

Variable	Total (n = 28)	Control Group (n = 14)	Switching Group (n = 14)	*p*
Age, years; mean ± SD	51.5 ± 8.7	50.4 ± 9.2	52.6 ± 8.4	0.51
Male, n (%)	15 (53.6%)	9 (64.3%)	6 (42.9%)	0.45
DKT, n (%)	17 (60.7%)	6 (42.9%)	11 (78.6%)	0.12
Transplant vintage, years; median (IQR)	3.3 (1.6–7.5)	3.2 (1.5–14.4)	3.3 (1.8–7.3)	0.85
Baseline serum creatinine, mg/dL; mean ± SD	1.34 ± 0.48	1.40 ± 0.59	1.27 ± 0.37	0.51
White blood cells, cells/μL; median (IQR)	5725 (5110–7110)	5710 (5280–6930)	5985 (4680–7290)	0.95
Neutrophil, cells/μL; median (IQR)	3720 (3145–4800)	3525 (3180–4270)	4055 (3120–5100)	0.73
Lymphocyte, cells/μL; median (IQR)	1645 (1290–2085)	1755 (1540–2100)	1455 (1180–2070)	0.43
Dosage of MMF, mg/day; mean ± SD	1179 ± 69	1250 ± 126	1107 ± 57	0.31
Tacrolimus trough level, ng/mL; mean ± SD	4.8 ± 1.0	4.7 ± 1.2	5.0 ± 0.8	0.59
Duration between first and second vaccinations, months; median (IQR)	2.8 (2.8–2.8)	2.8 (2.8–2.8)	2.8 (2.8–2.8)	0.14
Duration between second and third vaccinations, months; median (IQR)	1.3 (1.1–1.5)	1.3 (1.1–1.5)	1.3 (1.2–1.4)	0.72
Time since the last BNT162b2, months; median (IQR)	4.9 (4.6–5.3)	4.9 (4.6–5.3)	5.0 (4.8–5.2)	0.66
Baseline anti-SARS-CoV-2 S antibody, BAU/mL; median (IQR)	170.2 (36.0–510.3)	204.9 (44.7–541.2)	164.5 (19.5–429.5)	0.61

BAU, binding antibody units; DKT, deceased donor kidney transplantation; MMF, mycophenolate mofetil.

**Table 2 vaccines-10-01685-t002:** Adverse events related to the vaccination.

Adverse Events	Control Group (n = 14)	Switching Group (n = 14)
Immunosuppressant-related		
Oral ulcers, n (%)	0 (0%)	2 (1.4%)
Edema, n (%)	0 (0%)	0(0%)
Diarrhea, n (%)	0 (0%)	0 (0%)
Pneumonitis, n (%)	0 (0%)	0 (0%)
Rejection, n (%)	0 (0%)	0 (0%)
Vaccine-related		
Myalgia, n (%)	11 (78.6%)	13 (92.9%)
Fever, n (%)	3 (2.1%)	2 (1.4%)
Bleeding, n (%)	0 (0%)	0 (0%)
Chest discomfort, n (%)	0 (0%)	0 (0%)
Severe headache, n (%)	0 (0%)	0 (0%)
Vomiting, n (%)	0 (0%)	0 (0%)
Seizure, n (%)	0 (0%)	0 (0%)
Stroke-like symptoms, n (%)	0 (0%)	0 (0%)

## Data Availability

The data presented in this study are available on request from the corresponding author. The data are not publicly available due to ethical issue.

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
