# Peer review of "Benefits of Switching Mycophenolic Acid to Sirolimus on Serological Response after a SARS-CoV-2 Booster Dose among Kidney Transplant Recipients: A Pilot Study"

_vaccines, 2022, doi:10.3390/vaccines10101685_

Round 1

Reviewer 1 Report

The COVID pandemics is an ongoing important issue for human especially in the specific group with solid organ transplantation who are immunocompromised. This article discusses how to enhance the production of SARS-CoV-2 antibodies through shifting of immunosuppressants. I believe this may help these recipients to get more antibodies and higher antibody titers to prevent infection of COVID.

However, the most drawback is that we still don’t know the durability of this promising results. This will be more clinically significant for transplant society.

The second, what will happen after shifting back to previous immunosuppressants? Do you have any opinions or discussion on these two issues?

The 3rd, the check point of spike antibody is for these transplant patients who get the 3rd dose (booster dose for immunocompromised patients). Do you recommend to access immunity to SARS-CoV-2 following COVID-19 vaccination for these transplant patients since the FDA is not encouraging to do these tests for general population?

The fourth is that authors check antibodies at 14 days after vaccination. However, the serology expression may be sooner or later individualized. Therefore, positive test may provide a false sense of reassurance, on the contrary, negative test or lower titer may mislead the result which will possibly express the same titer later. 14 days may be not long enough for these immunocompromised patients.

The last, neutralizing antibody detection test is to detect the functional ability of antibodies. Could you explain a little bit of ElecsysR belong to which kind of neutralizing test (VNT, pVNT or cVNT) in context?

 In addition, lots of the reference format are not consistent.

Author Response

The COVID pandemics is an ongoing important issue for human especially in the specific group with solid organ transplantation who are immunocompromised. This article discusses how to enhance the production of SARS-CoV-2 antibodies through shifting of immunosuppressants. I believe this may help these recipients to get more antibodies and higher antibody titers to prevent infection of COVID.

However, the most drawback is that we still don’t know the durability of this promising results. This will be more clinically significant for transplant society.

Response: We agree with your recommendation. Unfortunately, we did not performed a longer follow-up antibody testing because we cannot exclude those with asymptomatic SARS-CoV-2 infection after this trial. We add this limitation in the last paragraph of page 9 of marked version.  

The second, what will happen after shifting back to previous immunosuppressants? Do you have any opinions or discussion on these two issues?

Response: After B-cell was activated for antibody production, switching the immunesuppression back to the standard regimen which mainly inhibit T-cells should not deplete antibody level in serum. We also added this discussion to the last paragraph of page 9 of marked version.

The 3rd, the check point of spike antibody is for these transplant patients who get the 3rd dose (booster dose for immunocompromised patients). Do you recommend to access immunity to SARS-CoV-2 following COVID-19 vaccination for these transplant patients since the FDA is not encouraging to do these tests for general population?

Response: As a pilot study, we still could not recommend antibody testing among kidney transplant recipients to see who are likely to gain benefits from drug switching. Larger RCT is needed to prove the benefits of switching immunosuppression especially seroconversion.  

The fourth is that authors check antibodies at 14 days after vaccination. However, the serology expression may be sooner or later individualized. Therefore, positive test may provide a false sense of reassurance, on the contrary, negative test or lower titer may mislead the result which will possibly express the same titer later. 14 days may be not long enough for these immunocompromised patients.

Response: We actually performed the antibodies test 4 weeks after vaccination as per page 3 (2.3. outcome measurement).

The last, neutralizing antibody detection test is to detect the functional ability of antibodies. Could you explain a little bit of ElecsysR belong to which kind of neutralizing test (VNT, pVNT or cVNT) in context?

Response: In this study, we did not performed a neutralizing test but we performed spike antibodies test.

 In addition, lots of the reference format are not consistent.

Response: Thank you. We re-formatted all references to MDPI style.

Reviewer 2 Report

To the authors

The study by Athiphat Banjongjit and co-authors addresses an important topic regarding our dilemma of a low seroresponse after vaccinations against COVID-19.

The authors hypothesized to improve vaccine efficacy against COVID 19 by switching from mycophenolic acid to sirolimus and low dose tacrolimus two weeks before until two weeks after the 4th vaccination.

The authors measured anti-SARS-CoV-2 S antibody levels 4 weeks after vaccination, a significantly bigger increase in the switch group was judged a better humoral response.

Major points:

-          Surprisingly, all SOTR had a positive ab titer right before the 4th vaccination (except 1), which is quite unusual in SOTR. (I would expect 40-60%.) Demonstrating seroconversion in a former seronegative patient after switch of immunosuppression would have proven effectiveness much more clearly than just a bigger rise of antibody titers. Moreover, there is little evidence that higher antibody titers predict a better humoral response and, as a consequence, better protection against infection.

-          Thus, how do the authors explain the almost 100% response rate, and what is the rationale and the need to switch immunosuppression in this cohort?

-          It is not clear to me, why 33 from 64 screened patients were excluded from the study. How many SOTR are seen in the department?  Regarding Figure 1, please specify the reasons for exclusion including the number of patients.

-          Did the authors exclude recent or previous SARS-CoV-2 infection before inclusion of the study until 4 weeks after vaccination, and if yes, how? Regular swab tests? At a minimum, measurements of SARS-CoV-2 N protein should have been performed and mentioned in the manuscript.

-          The differences between both groups should be characterized in more detail: dosage of MMF, trough levels of tacrolimus before switch, trough levels reached with the use of mTORi (before 4th vaccination and levels up to 2 weeks after), and trough levels of tacrolimus during the same period in both groups. How often were measurents of tac and mTORi levels performed? If, for instance, mTORi drug levels remained very low, it might be comparable to regimens in which MMF is omitted without substitution to another immunosuppressive drug.

-          The rationale to also lower tac trough levels in the switch group is not discussed. How do authors differentiate the benefit of switching to mTORi rather than reduction of tac doses?

-          Were there changes in creatinine and albuminuria during the study? This should be mentioned within the result section.

-          Line 213: The term “superior immune response” is wrong and misleading (see above), especially due to missing B and T cell responses (e.g. IGRA for T cell responses).

-          Line 227-228 and line 233-234:  “A four-week period of switching immunosuppression which was two weeks before and two weeks after vaccination of the present study is sufficient to improve antibody response” and  “the two-week duration of immunosuppressive regimen switching prior vaccination to allow immune restoration and another two weeks after vaccination for vaccine response in the present study is appropriate” -> cannot be drawn as a conclusion from that pilot study and should be interpreted with more caution.

Minor points:

-          line 48: add reference and numbers (percentage) of high mortality rate

-          Line 49: add reference and numbers of decrease of mortality over time

-          Line 76 the term “randomized controlled trial” should be deleted from the title and from the method section, since there were too few patients in each group to justify relevant conclusions for that pilot study

-          Line 85: “having a history of SARS-CoV-2 infection” -> please add information, how this was confirmed (positive PCR swab test? N protein of SARS-CoV-2?)

-          Line 86: what about DSA positivity?

-          Figure 1: please add another horizontal line with the positive (reactive) antibody level

-          Figure 1: why -5000? Where do results below 0 come from?

-          the impact of switching to mTOR inhibitor might be less relevant in stable SOTR without history of rejection or DSA. However, this strategy might help in patients with a history of rejection, repeated transplantation or higher immunological risk. I would suggest, to add this in the discussion.

-          Line 237 to 248: This part of the discussion is not relevant for the study and misses the context, it should be removed

-          Limitations of the study should be emphasized more clearly: small numbers, almost 100% positive antibody response before vaccination, assessment of anti-SARS-CoV-2 S antibody levels as the only marker

-          Please state the time intervals between 1st , 2nd and 3rd vaccination in both groups.

Author Response

To the authors

The study by Athiphat Banjongjit and co-authors addresses an important topic regarding our dilemma of a low seroresponse after vaccinations against COVID-19.

The authors hypothesized to improve vaccine efficacy against COVID 19 by switching from mycophenolic acid to sirolimus and low dose tacrolimus two weeks before until two weeks after the 4th vaccination.

The authors measured anti-SARS-CoV-2 S antibody levels 4 weeks after vaccination, a significantly bigger increase in the switch group was judged a better humoral response.

Major points:

-          Surprisingly, all SOTR had a positive ab titer right before the 4th vaccination (except 1), which is quite unusual in SOTR. (I would expect 40-60%.) Demonstrating seroconversion in a former seronegative patient after switch of immunosuppression would have proven effectiveness much more clearly than just a bigger rise of antibody titers. Moreover, there is little evidence that higher antibody titers predict a better humoral response and, as a consequence, better protection against infection.

-          Thus, how do the authors explain the almost 100% response rate, and what is the rationale and the need to switch immunosuppression in this cohort?

Response: Nearly 100% seropositive SOTRs might be resulted from small sample sizes. We add the suggestion to assess seroconversion rate in future study in the last paragraph of page 9 of marked version. Previous studies showed that higher anti-spike antibodies levels were correlated with neutralizing antibodies levels (10.1002/jmv.27287), and vaccine efficacy (10.1038/s41591-021-01540-1). Thus, we assumed that achieving higher anti-spike antibodies levels resulted in protection from symptomatic or at least severe infection. We already mentioned this limitation and suggested for future study about the vaccination efficacy in the last paragraph of page 9 of marked version.

-          It is not clear to me, why 33 from 64 screened patients were excluded from the study. How many SOTR are seen in the department?  Regarding Figure 1, please specify the reasons for exclusion including the number of patients.

Response: We initially included the KTRs who received two doses of AZD1222 and a single dose of BNT162b2 that was 64 patients from 693 active KTRs in our center. Then we exclude these patients as per new figure 1.

-          Did the authors exclude recent or previous SARS-CoV-2 infection before inclusion of the study until 4 weeks after vaccination, and if yes, how? Regular swab tests? At a minimum, measurements of SARS-CoV-2 N protein should have been performed and mentioned in the manuscript.

Response: Thank you for your suggestion. We exclude the SARS-CoV-2 infection by swab the symptomatic patients. Unfortunately, we cannot perform anti-N protein test. We agree that this is the major limitation and we add this limitation in the last paragraph of page 10. 

-          The differences between both groups should be characterized in more detail: dosage of MMF, trough levels of tacrolimus before switch, trough levels reached with the use of mTORi (before 4th vaccination and levels up to 2 weeks after), and trough levels of tacrolimus during the same period in both groups. How often were measurents of tac and mTORi levels performed? If, for instance, mTORi drug levels remained very low, it might be comparable to regimens in which MMF is omitted without substitution to another immunosuppressive drug.

Response: We add the baseline dosage of MMF, TAC trough level in table 1. The trough level of TAC was added in page 3 “In the switching group, a trough level achieved 3.3 ±1.6 ng/mL for TAC and 10.3 ±4.0 ng/mL for sirolimus”. The levels of TAC was performed at baseline in both group. The levels of TAC and mTORi were performed 2 weeks after sirolimus was administered in switching group. We did not measured TAC level in control group at the same period because there was no dosage adjustment. We added these details in page 3 in subject 2.2. Immunosuppressive regimens and vaccination  

-          The rationale to also lower tac trough levels in the switch group is not discussed. How do authors differentiate the benefit of switching to mTORi rather than reduction of tac doses?

Response: Thank you for your comment. The better immune response in the mTORi group can be explained by both replacement of MPA by mTORi and reduction in Tac level.   We added this in the discussion part.

-          Were there changes in creatinine and albuminuria during the study? This should be mentioned within the result section.

Response: Thank you for your recommendation. We added these two non-significant change outcomes in the adverse events (page 6).

-          Line 213: The term “superior immune response” is wrong and misleading (see above), especially due to missing B and T cell responses (e.g. IGRA for T cell responses).

Response: Thank you. We replace the term “superior immune response” with “higher antibody level” in page 8.

-          Line 227-228 and line 233-234:  “A four-week period of switching immunosuppression which was two weeks before and two weeks after vaccination of the present study is sufficient to improve antibody response” and  “the two-week duration of immunosuppressive regimen switching prior vaccination to allow immune restoration and another two weeks after vaccination for vaccine response in the present study is appropriate” -> cannot be drawn as a conclusion from that pilot study and should be interpreted with more caution.

Response: Thank you for giving the crucial comment.   We rewrite the discussion to understate the our conclusion.

Minor points:

-          line 48: add reference and numbers (percentage) of high mortality rate

Response: We used the same reference and numbers as line 49 (reference 1).

-          Line 49: add reference and numbers of decrease of mortality over time

Response: We add the percentage of mortality rate over time as your suggestion.

-          Line 76 the term “randomized controlled trial” should be deleted from the title and from the method section, since there were too few patients in each group to justify relevant conclusions for that pilot study

Response: We change the term “randomized controlled trial” to “pilot study” in the title, abstract and method as your suggestion.

-          Line 85: “having a history of SARS-CoV-2 infection” -> please add information, how this was confirmed (positive PCR swab test? N protein of SARS-CoV-2?)

Response: We detailed this sentences with “(had a positive SARS-CoV-2 polymerase chain reaction [PCR] result),”

-          Line 86: what about DSA positivity?

Response: We didn’t enroll the patient with positive DSA.  We added the exclusion in the method part.

-          Figure 1: please add another horizontal line with the positive (reactive) antibody level

Response: Thank you. We already had this horizontal line and we add “The horizontal dotted line indicated the cutoff point at 0.823 BAU/mL.” in the legend.

-          Figure 1: why -5000? Where do results below 0 come from?

Response: The lowest point in violin plot do not represent the lowest value of the population but it showed the probability instead. 

-          the impact of switching to mTOR inhibitor might be less relevant in stable SOTR without history of rejection or DSA. However, this strategy might help in patients with a history of rejection, repeated transplantation or higher immunological risk. I would suggest, to add this in the discussion.

Response: Patients enrolled in this study were relatively low risk for rejection.  To apply this finding in the higher risk patients, we may need more study.   We added this concern in the limitation part.

-          Line 237 to 248: This part of the discussion is not relevant for the study and misses the context, it should be removed

Response: We removed this paragraph as your suggestion.

-          Limitations of the study should be emphasized more clearly: small numbers, almost 100% positive antibody response before vaccination, assessment of anti-SARS-CoV-2 S antibody levels as the only marker

Response: Thank you. We added these limitations as your suggestion.

-          Please state the time intervals between 1st , 2nd and 3rd vaccination in both groups.

Response: We added these interval in Table 1.

Reviewer 3 Report

Interesting paper about mTORi-based regimen to improve response to SARS-CoV2 vaccination. Some points limit the generalizability and should be discussed.

- Vaccination is performed in every patient with a viral agent with a subsequent booster dose with an mRNA vaccine. Please stress this aspect in the title and discussion (the worldwide approach could be different, considering that transplanted patients were preferentially driven to mRNA products for better response and tolerance).

- mTORi have potentially positive effects on viral infections but without unique outcomes for every condition (see 10.1111/tid.12601); however, rapid and repeated switching to therapy exposes potential under-immunosuppression with an associated risk of rejection. These results should be more extensively discussed, even considering a longer f/up for adverse events (see also other points of the rev report)

- cellular response to vaccination could be present instead of the increase in IgG anti-SARS-CoV2; if not tested, it should be stressed as a limitation in the results and discussion

- the increase of IgG should be present but could be lessened after re-switching considering a longer f/up; please extend the f/up for adverse events or re-control of anti-SARS-COV2 IgG and discuss these data (if available or as an important limitation if not)

- please include in the introduction/discussion the rate of infection in your area and transplant experience, and adopted therapy in case of SARS-COV2 clinical disease (also discussing possible interference/improvement of the adopted regimen or mTORi with other treatments such as IL-6 blocking drugs, see 10.1111/tid.13348).

- Table 2 reported oral ulcers in the control group, but I think that, as described in the text, these events were observed in the switching group. Please reconcile.

Author Response

Interesting paper about mTORi-based regimen to improve response to SARS-CoV2 vaccination. Some points limit the generalizability and should be discussed.

- Vaccination is performed in every patient with a viral agent with a subsequent booster dose with an mRNA vaccine. Please stress this aspect in the title and discussion (the worldwide approach could be different, considering that transplanted patients were preferentially driven to mRNA products for better response and tolerance).

Response: Thank you for your concern.   We focused on the immune response during second mRNA vaccination.   Since there was a worldwide shortage of mRNA at the first launch of vaccine.   Many countries decided to choose primary regimen with viral vector vaccine such as UK, Korea, Australia, and many countries in Asia.   We believe that our study has some benefit in these countries.   And the concept of switching the regimen of immunosuppression can be applied to all vaccination regimens.

- mTORi have potentially positive effects on viral infections but without unique outcomes for every condition (see 10.1111/tid.12601); however, rapid and repeated switching to therapy exposes potential under-immunosuppression with an associated risk of rejection. These results should be more extensively discussed, even considering a longer f/up for adverse events (see also other points of the rev report)

Response: Thank you for your concerns.  We agree with you that longer term follow up is needed to ensure the safety of the switching regimen.   However, the regimen we used in this study has been proved in many studies for regarding efficacy and safety such as TRANSFORM study and ATHENA study.   We added this concern in the limitation of the study.  

- cellular response to vaccination could be present instead of the increase in IgG anti-SARS-CoV2; if not tested, it should be stressed as a limitation in the results and discussion

Response: We added this limitation in page 10.

- the increase of IgG should be present but could be lessened after re-switching considering a longer f/up; please extend the f/up for adverse events or re-control of anti-SARS-COV2 IgG and discuss these data (if available or as an important limitation if not)

Response: We changed the duration of the study up to September 2022 (1.1 study design), added “The adverse events were monitored for six months until September 2022.” in 2.3. outcome measurements. The rejection rate which was 0% in both groups was added in Table 2. Other adverse events were at the same rate. However, we did not performed a longer follow-up antibody testing. We add this limitation in the last paragraph of page 9 in marked version.  

- please include in the introduction/discussion the rate of infection in your area and transplant experience, and adopted therapy in case of SARS-COV2 clinical disease (also discussing possible interference/improvement of the adopted regimen or mTORi with other treatments such as IL-6 blocking drugs, see 10.1111/tid.13348).

Response: We added the infection rate and severity among KTRs in our center in the introduction. We did not assessed switching the mTORi during the infection; thus we did not mentioned the interference of mTORi with other treatment.

- Table 2 reported oral ulcers in the control group, but I think that, as described in the text, these events were observed in the switching group. Please reconcile.

Response: Thank you very much for this point. We did a mistake switching the rate of oral ulcer between control and switching group in Table 2 and we already corrected the data.

Round 2

Reviewer 1 Report

The last question is that the switching group, a trough level achieved 3.3 ±1.6 ng/mL for TAC and 10.3 ±4.0 ng/mL for sirolimus, which is relative higher than the usual maintenance sirolimus drug level. In addition, the adjusted [Tac]  and add on [Sir] had better be prensented on the table 1 also. In fact, These patinets may be still under over immunosuppression. Further reduction of sirolimus (under 2 mg) may be beneficial to the antibody production further. I don`t know why you have to keep a higher maintenance [Sir] in your series as compared with general maintenance trough level. 

Author Response

The last question is that the switching group, a trough level achieved 3.3 ±1.6 ng/mL for TAC and 10.3 ±4.0 ng/mL for sirolimus, which is relative higher than the usual maintenance sirolimus drug level. In addition, the adjusted [Tac]  and add on [Sir] had better be prensented on the table 1 also. In fact, These patinets may be still under over immunosuppression. Further reduction of sirolimus (under 2 mg) may be beneficial to the antibody production further. I don`t know why you have to keep a higher maintenance [Sir] in your series as compared with general maintenance trough level. 

Response: Thank you for your suggestion.

It was a short period of time after switching, so the sirolimus dose adjustment was achieved high normal. However, after the trough level was measured, we adjust the dosage to aim 6-10 ng/mL.  

Because adjusted [Tac] and [Sir] were presented only among switching group, we did not put this data in the table 1.    

Reviewer 2 Report

Dear authors,

thank you very much for sending a revised version of your manuscript with a lot of changes made.

The revised version has now improved.

Minor points:

Sentence in the last paragraph:

"Nevertheless, after the B-cell was activated and could produce antibodies, switching the MPA sparing immunosuppressive regimen back to the standard regimen should not deplete antibody level in the serum because it mainly inhibits the production of T cells"

Please delete this sentence, because it is not this simple and I advice not to draw conclusions like this without detailed experimental B and T cell responses.

Author Response

Minor points:

Sentence in the last paragraph:

"Nevertheless, after the B-cell was activated and could produce antibodies, switching the MPA sparing immunosuppressive regimen back to the standard regimen should not deplete antibody level in the serum because it mainly inhibits the production of T cells"

Please delete this sentence, because it is not this simple and I advice not to draw conclusions like this without detailed experimental B and T cell responses.

Response: Thank you for your recommendation. We remove this paragraph as your suggestion.

Reviewer 3 Report

The authors performed all suggested revisions. No further comments are needed in my opinion.

Author Response

The authors performed all suggested revisions. No further comments are needed in my opinion.

Response: Thank you.